# Influence of Sociodemographic and Emotional Factors on the Relationship between Self-Compassion and Perceived Stress among Men Residing in Brazil during the COVID-19 Pandemic

**DOI:** 10.3390/ijerph19138159

**Published:** 2022-07-03

**Authors:** Emanuel Missias Silva Palma, Anderson Reis de Sousa, Jules Ramon Brito Teixeira, Wanderson Carneiro Moreira, Ana Caroline Monteiro de Araújo, Luiz Filipe Vieira Souza, Júlio Cézar Ramos dos Anjos, Hannah Souza de Almeida Portela, Herica Emilia Félix de Carvalho, Vinícius de Oliveira Muniz, Nilo Manoel Pereira Vieira Barreto, Éric Santos Almeida, Tilson Nunes Mota, Sélton Diniz dos Santos, Antônio Tiago da Silva Souza, Josielson Costa da Silva, Camila Aparecida Pinheiro Landim Almeida, Luciano Garcia Lourenção, Aline Macêdo de Queiroz, Edmar José Fortes Júnior, Magno Conceição das Merces, Shirley Verônica Melo Almeida Lima, Francisca Michelle Duarte da Silva, Nadirlene Pereira Gomes, Maria Lúcia Silva Servo, Evanilda Souza de Santana Carvalho, Sônia Barros, Tânia Maria De Araújo, Márcia Aparecida Ferreira de Oliveira, Álvaro Francisco Lopes de Sousa, Isabel Amélia Costa Mendes

**Affiliations:** 1Psychology Course, Escola Bahiana de Medicina e Saúde Pública, Salvador 40231-300, BA, Brazil; emanuelmssilva@gmail.com (E.M.S.P.); monteiroacaroline@gmail.com (A.C.M.d.A.); 2College of Nursing, Universidade Federal da Bahia (UFBA), Salvador 40231-300, BA, Brazil; luiz_filipe011@hotmail.com (L.F.V.S.); ceu200992@gmail.com (J.C.R.d.A.); hannahportelaenfermeira@gmail.com (H.S.d.A.P.); nilo.manoel@ufba.br (N.M.P.V.B.); eriksdn@gmail.com (É.S.A.); josielson.silva@ufba.br (J.C.d.S.); nadirlenegomes@hotmail.com (N.P.G.); 3Health Department, Universidade Estadual de Feira de Santana (UEFS), Feira de Santana 44001-970, BA, Brazil; julesramon@gmail.com (J.R.B.T.); selton.diniz@yahoo.com.br (S.D.d.S.); mlsservo@uefs.br (M.L.S.S.); evasscarvalho@uefs.br (E.S.d.S.C.); araujo.tania@uefs.br (T.M.D.A.); 4College of Nursing, Universidade de São Paulo (USP), Sao Paulo 05403-000, SP, Brazil; wanderson.moreira@usp.br (W.C.M.); sobarros@usp.br (S.B.); marciaap@usp.br (M.A.F.d.O.); 5Ribeirao Preto College of Nursing, Universidade de São Paulo (USP), Ribeirao Preto 14040-902, SP, Brazil; hericacarvalho@usp.br (H.E.F.d.C.); sousa.alvaromd@gmail.com (Á.F.L.d.S.); iamendes@usp.br (I.A.C.M.); 6Nursing Course, Faculty Doctum, Serra 29168-064, ES, Brazil; viniciusomuniz22@gmail.com; 7Board of Health Surveillance, Secretaria de Saúde do Estado da Bahia (SESAB), Salvador 40130-160, BA, Brazil; tilson.nunes.mota@gmail.com; 8Department of Medicine, Universidade Federal do Delta do Parnaíba (UFDPar), Parnaiba 64215-343, PI, Brazil; antoniotiago84@gmail.com; 9Institute of Health Sciences, Universidade Católica Portuguesa (UCP), 4200-072 Porto, Portugal; camilaapapila@hotmail.com; 10School of Nursing, Universidade Federal do Rio Grande (FURG), Rio Grande 96201-900, RS, Brazil; lucianolourencao.enf@gmail.com; 11Faculty of Nursing, Universidade Federal do Pará (UFPA), Belem 66075-110, PA, Brazil; alinemacedo@ufpa.br; 12Medicine Course, Instituto de Educação Superior do Vale do Parnaíba (IESVAP), Parnaiba 64215-343, PI, Brazil; edmarfortes@hotmail.com; 13Department of Life Sciences, Universidade do Estado da Bahia (UNEB), Salvador 41150-000, BA, Brazil; mmerces@uneb.br; 14Department of Nursing, Universidade Federal de Sergipe (UFS), Lagarto 49400-000, SE, Brazil; shirleymelo.lima@gmail.com; 15Psychology Course, Instituto Federal de Educação, Ciência e Tenologia do Maranhão (IFMA), Sao Luís 65068-669, MA, Brazil; michelleduartes@yahoo.com.br; 16Graduate Program in Nursing, Health Sciences Center, Universidade Federal de Santa Maria (UFSM), Santa Maria 97105-900, RS, Brazil; 17Global Health and Tropical Medicine (GHTM), Instituto de Higiene e Medicina Tropical, Universidade Nova de Lisboa, 1349-008 Lisbon, Portugal

**Keywords:** mental health, men’s health, compassion, psychological stress, COVID-19

## Abstract

The analysis of sociodemographic and emotional factors is essential to understanding how men perceive stress and practice self-compassion. In health crises, this problem becomes an emergency for public health. This study aimed to analyze the influence of sociodemographic and emotional factors on the relationship between self-compassion and the perceived stress of men residing in Brazil during the COVID-19 pandemic. This is a nationwide cross-sectional study carried out between June and December 2020 with 1006 men who completed a semi-structured electronic questionnaire. Data were collected using the snowball technique. Perceived stress was measured by the Perceived Stress Scale (PSS-14), and self-compassion was assessed using the Self-Compassion Scale. Most men had low self-compassion (51.5%; *n* = 516) and a moderate level of perceived stress (60.9%; *n* = 613), while 15.9% (*n* = 170) had a high level of stress. The prevalence of men in the combined situation of low self-compassion and high perceived stress was 39.4% (*n* = 334). Living with friends had a higher prevalence of low self-compassion and high perceived stress. The prevalence of common mental disorders was high (54.3%). Men with low levels of self-compassion reported higher levels of perceived stress; however, this association was moderated by emotional and sociodemographic variables. These findings highlight the importance of considering individual and contextual factors in public policies promoting men’s mental health.

## 1. Introduction

The Coronavirus Disease 2019 (COVID-19) pandemic increased reports of emotional and behavioral problems across different population groups worldwide. Indicators of depression, anxiety, anger, fear, irritability, and substance use have been consistently identified [1,2,3,4,5,6,7,8,9], especially in adult men [10,11].

Although, to date, no consensus has been reached on the determinants of higher mortality from COVID-19 among the male population, factors associated with lifestyle, such as smoking and alcoholism, and delay in seeking health services may be involved [12]. In addition, the male population is less likely to adhere to COVID-19 prevention measures and be more involved in health risk behaviors than women [13,14]. Men are more likely to participate in large public meetings or have close physical contact with others [15,16,17]. They also tend to have lower rates of handwashing, social distancing, and wearing masks [15,18,19,20]. Finally, men have presented worryingly high levels of common mental disorders (CMD) and perceived stress during the COVID-19 pandemic [21,22,23]. 

Mental health problems have demanded increasing attention from the literature, especially stress. Perceived stress refers to the self-assessment of the discrepancy between the demands imposed by the context (i.e., stressors) and the availability of personal resources to deal with such demands effectively [24]. In other words, perceived stress involves an individual’s appraisal of the degree to which environmental variables are uncontrollable and unpredictable, requiring personal efforts to mitigate negative outcomes [24]. In this sense, perceived stress has been considered a psychosocial vulnerability factor, leading to adverse outcomes in mental and physical health [21,24]. For instance, in studying the role of perceived stress in acute coronary syndrome, literature findings indicate that men were more likely to be impacted by stress than women [25]. In addition, significant increases in stress levels due to the COVID-19 pandemic have been observed in different countries, which may pose risks of developing more severe disorders for various groups [26].

Taken together, these study findings highlight the need to consider the deleterious effects of stress on health and the protective role that individual and contextual variables may play in buffering these consequences. Consequently, there has been an urgent call for studies and interventions that help understand the intricate relationships between risk and protective factors and mental health outcomes in the COVID-19 pandemic context. 

Self-compassion is an individual-level protective factor that can be formally trained or consistently identified in the population [27]. Therefore, it has been considered a promising intervention target at individual and collective levels and sparked the growing interest of researchers from different fields [28]. Self-compassion is an attitude of openness to one’s suffering in an understanding, welcoming, and non-judgmental way [28]. These authors highlighted three constitutive dimensions of self-compassion: (a) acting in an understanding and caring way towards oneself; (b) accepting everyday difficulties as natural occurrences of human life; and (c) adopting a rational and conscious posture in the face of personal suffering. 

The literature has shown positive associations between self-compassion and favorable mental health outcomes [27,29,30,31]. Notably, studies carried out in the context of the COVID-19 pandemic indicate that self-compassion plays a critical protective role in the mental health of different populations [32,33]. For instance, Gutiérrez-Hernández et al. (2021) [34] observed that self-compassion was associated with lower emotional problems during the COVID-19 lockdown in Spain. Similarly, a study with a Turkish sample found that higher levels of self-compassion were associated with less intolerance of uncertainty and fear of COVID-19 and greater well-being [35]. These results highlight the direct role that self-compassion might play in protecting individuals’ mental health during the pandemic. However, gaps remain regarding sociodemographic and psychosocial factors in the relationship between self-compassion and mental health outcomes, such as perceived stress. In the present study, we addressed these issues by investigating the intermediary roles that some intra-individual and contextual factors play in this relationship. 

It is essential to highlight that more thorough investigations are encouraged to better understand the relationships between the determinants of psychological distress (e.g., perceived distress) and the potential protective factors (e.g., self-compassion). Furthermore, it is crucial to advance the understanding of the factors that, when affecting these relationships, potentiate or weaken their effects. Therefore, we emphasize that sociodemographic factors such as gender, education, and age, and the simultaneous existence of indicators of psychological distress (i.e., common mental disorders—CMD) might affect the relationship between self-compassion and perceived stress. In this sense, the present study aimed to analyze the influence of sociodemographic and emotional factors on the relationship between self-compassion and the perceived stress of men residing in Brazil during the COVID-19 pandemic.

## 2. Method

### 2.1. Type of Study

This is a cross-sectional, nationwide, online survey. The research study occurred between March and May 2020, during the critical period of social distancing determined by the Brazilian health authorities due to the COVID-19 pandemic.

### 2.2. Population, Sample, and Eligibility Criteria

For the sample size calculation, the following parameters were considered: a population of 64,520,660 Brazilian men with internet access [36]; the expected prevalence of the outcome of 50%; 95% confidence level; 5% accuracy; 80% power; study design effect of 2; and a 20% addition for losses. Thus, the sample was estimated at 923 participants.

Inclusion criteria were: age over 18 years and residing in Brazil during the data collection period. In addition, participants who declared themselves with non-binary gender identities were excluded from the study. 

### 2.3. Data Collection Instrument

A structured online questionnaire was used for data collection. Methodological procedures were adopted to construct the questionnaire, including validation by the group of researchers and 20 men from the study population (pilot study) for calibration and content analysis purposes. Therefore, data from these participants were not included in the final sample analyzed.

The form was organized into thematic blocks, including the following variables of interest: (1) sociodemographic data (sexual identity, gender, age, schooling, race/skin color, marital status, monthly income, occupation, housemates, and use of the health system) and (2) the psychosocial and emotional aspects, including the evaluation of CMD, self-compassion, and perceived stress.

In Brazil, the concepts of race/skin color in official institutions and documents admit options for: Black: very dark-skinned; White: Light-skinned in appearance; Yellow: Asians (Japanese, Chinese, and Korean); Brown: lighter-skinned (children of white and black, indigenous and black, indigenous and white); and Indigenous: Descendants of Brazilian Indians based on self-declaration [37]. The variable measurement took place through this list of options defined by the Instituto Brasileiro de Geografia e Estatística (Brazilian Institute of Geography and Statistics—IBGE). The participants were instructed to identify themselves with one of the options and answer: How do you recognize your race/skin color? Regarding income, in 2021, the minimum wage in Brazil was US$21,489 [38].

Sexual orientation was categorized into heterosexual and non-heterosexual (homosexual, bisexual, pansexual, asexual, and others). Gender was dichotomous into cisgender and non-cisgender (transgender, non-binary person, and male trans person) [39]. The participants were asked: How do you recognize your sexual orientation? How do you recognize your gender identity? To fill in these variables, the participants received information on defining a person with a cisgender identity and a transgender person—a trans man and a trans male person. Regarding employment status, the participants who had a formal contract or were civil servants were classified as having a formal employment status. Other conditions were classified as informal.

The psychosocial and emotional variables were evaluated using the following instruments:

Self-Reporting Questionnaire 20 (SRQ-20) is an instrument proposed by the World Health Organization to screen CMD in developing countries. The SRQ-20 assesses neurotic symptomatology, including symptoms of anxiety, depression, loss of vital energy, mood, and psychosomatic disorders [40]. It contains 20 questions about how the individual felt in the last four weeks (e.g., Are you easily frightened? and Do you cry more than usual?) and is answered using a dichotomous scale (yes = 1 or no = 0). It was validated in Brazil and demonstrated good psychometric properties (its internal consistency was α = 0.88) [40,41,42]. The cut-off point for suspected CMD was ≥5, as recommended in studies in Brazil [41].

The Perceived Stress Scale (PSS-14) [24] is a self-report instrument used to measure levels of perceived stress (e.g., In the last month, how often have you been upset because of something that happened unexpectedly? Or, In the last month, how often have you been angered because of things that happened that were outside of your control?) during the previous four weeks. The response options use a five-point Likert scale ranging from 0 = never to 4 = very often. The scale was validated in the Brazilian context [43] and showed good evidence of validity and reliability (internal consistency was α = 0.88). Stress levels were dichotomized into low (cut-off point ≤ 26) and high (moderate/high, cut-off point ≥ 27).

The Self-Compassion Scale [44] is the self-report instrument used to measure self-compassion. It consists of 26 items (e.g., “I’m disapproving and judgmental about my flaws and inadequacies”, or, “When something painful happens, I try to take a balanced view of the situation”) that are evaluated using a five-point Likert scale where 1 = almost never and 5 = almost always. It was adapted and validated in the Brazilian context and presented good validity evidence with an internal consistency of α = 0.92 [45]. Self-compassion levels were dichotomized into low (cut-off point ≤ 82) and high (cut-off point ≥ 83).

### 2.4. Data Collection Procedures

After the instruments were calibrated, the sample definition and data collection strategy were carried out. For this, the consecutive recruitment technique called snowball was used. In this sense, we gathered a group of 25 participants, five from each region of the country, accessed through the digital social networks of Facebook^®^ Instagram^®^, WhatsApp^®^ (Menlo Park, CA, USA), Twitter^®^ (San Francisco, CA, USA) Telegram^®^, relationship applications—for access by men who have sex with other men such as Grindr^®^ (West Hollywood, CA, USA), Hornet^®^—Queer Social Network (Los Angeles, CA, USA), Scruff^®^ (New York, United States) and Tinder^®^ (West Hollywood, Los Angeles, CA, USA) and e-mails available on public websites, such as companies, schools, universities, and Brazilian public departments. Sampling through the snowball strategy may have limited the reach of variables, such as race/color, social class, age group, and gender identity, which is evidenced in populations of indigenous, poor, non-heterosexual, and transgender men. To reduce the risk of bias, the initially invited participants were selected in a way that ensured the representativeness of these population groups. 

The first five intentional sample groups configured the seeds in which the participants, in addition to being invited to participate in the study, were encouraged to forward the research invitation to other participants in their contact network. This methodological procedure made it possible to reach 27 seeds and their respective fruits, thus reaching all Brazilian states. It is worth noting that, when conducting a survey involving big data, the number of participants who had access to the survey invitation cannot be measured. However, it was possible to register eight refusals (0.7%).

In order to get closer to the participants, a planned data collection strategy was developed, namely: the use of a social network linked to the research group that conducted the study, through a page on Instagram named @cuidadoasaudehomens, through the elaboration of a personalized layout of the research, through the creation of digital animation to send the invitation and interaction with participants in the virtual environment, holding meetings and conversation circles, and a presentation of the research in virtual spaces of male socialization, which were all used as a way of propagating the research in different areas and capturing other groups of men residing in the country. This page was created by marketing, design, and communication professionals and is used for recruitment, approaching research participants, disseminating research results, and posting guidelines for health promotion. 

All contacts with participants were made individually to ensure participants’ privacy and the confidentiality of the data collected, which follows the general law of data protection and the resolution on ethics in research in a virtual environment.

The research used a self-administered, semi-structured electronic form composed of structured questions related to sociodemographic, occupational, and health characteristics and aimed at indicating the experiences of the male participants during the pandemic. Open questions were also inserted; however, they were not incorporated in this study. The Google Forms^®^ platform was used, which is free to use, as it is a widely-known platform in Brazil, is easy to access and use, and uses resources to ensure data security and protection, such as encryption.

### 2.5. Data Processing and Analysis

After collection, procedures were performed to verify the quality and eligibility of the data. First, the incompletely filled and repeated or duplicated forms were analyzed and excluded from the final sample. Then, data were treated and processed using Stata software version 16.1 (Stata Corp, College Station, TX, USA), version 14.0. First, The Kolmogorov–Smirnov test was used to assess the normality of the distribution of variables. Then, descriptive analyses were performed with the estimation of absolute and relative frequencies (categorical variables), average, standard deviation (SD), and minimum and maximum values (quantitative variables). Finally, prevalence, prevalence ratios (PR), and their respective confidence intervals (95%CI) were estimated for the outcome of interest (combined situation between low self-compassion and high level of perceived stress) and the strata of independent variables.

Using Pearson’s chi-square test, bivariate analyses were conducted to test the association between the outcome and the independent variables. Variables that obtained a *p*-value ≤ 0.20 were included in the multivariate analysis. Poisson regression was used with robust variation in the multivariate model using the backward procedure. The lowest value of the Akaike Information Criterion (AIC) was considered to select the best final model and the Variance Inflation Factor (VIF) for the diagnosis of multicollinearity (individual and medium VIF < 10). The statistical significance level of the tests was 5% (*p*-value ≤ 0.05).

### 2.6. Ethical and Legal Aspects

The study complied with ethical precepts, respecting the regulations enforced in Brazil for conducting research involving human beings [46,47]. The Research Ethics Committee approved the project—report number: 4.087.6111. All participants signed the consent form online.

## 3. Results

A total of 1006 Brazilian men participated in the study, with a predominance of non-heterosexual (54.1%) and cisgender (93.2%) men between 18- and 39 years old (76.3%), with a university degree (73.8%), of non-black skin (80.4%), without a partner (67.3%), with a monthly income of up to four minimum wages (64.2%), with a formal work situation (53.3%), and living with family members (69.8%) (Table 1). This profile presents some similarities with the profile of the Brazilian male population, composed predominantly of cisgender individuals aged 18 to 49 and with non-black skin. However, the sample has some differences concerning the sociodemographic profile of the population, such as higher education level and a higher income than the average Brazilian.

It was identified that most men had low self-compassion (516; 51.5%). Men with a moderate level of perceived stress predominated (613; 60.9%), followed by low-level (223; 22.2%) and high-level (170; 15.9%). Almost half of the men (48.6%) were exposed to a combined high perception of stress and low self-compassion, indicating an unfavorable and worrying condition. The prevalence of suspected CMD was high—i.e., 54.3% (Table 1). 

In the bivariate analysis, the association between the combined situation of low self-compassion and high perceived stress, at statistical levels, was observed for the variables of gender (cisgender: PR = 1.32), marital status (no partner: PR = 1.22), for those who shared a house with a friend (PR = 1.27), and men with CMD (PR = 2.82) (Table 2). These results indicate that the association under analysis (low self-compassion and high perceived stress) varies according to the strata of these variables.

In the multivariate modeling, the variables of gender, age group, education, marital status, monthly income, and use of the health system did not obtain a statistically significant association, and their exclusion, individually and sequentially, from the highest to the lowest *p*-value, was endorsed by the decrease in the value of AIC.

The associations between low self-compassion and high perceived stress in Brazilian men remained statistically significant for the following conditions: living with friends (PR: 1.26; 95%CI 1.03–1.53) and CMD (PR: 2.83; 95%CI: 2.39–3.36) (Table 3). This final model had the lowest AIC value (1.57) and a medium VIF of 1.15 (data not shown in tables).

## 4. Discussion

This study set out to analyze the influence of sociodemographic and emotional factors on the relationship between self-compassion and the perceived stress of men residing in Brazil during the COVID-19 pandemic. The experience of a combined situation of high perceived stress and low self-compassion was very significant in the male population investigated (reaching approximately half of the participants). The situation shows an apparent demand for attention, care, and coping actions as it poses a significant problem for public policies. 

Based on the bivariate analysis results, a combined situation between low self-compassion and high-to-moderate levels of perceived stress was more frequent among non-cisgender, single Brazilian men living with friends or alone and having a suspected common mental disorder. 

In this sense, it is noteworthy to highlight that self-compassion has been considered a protective factor for mental health and is associated with skillful emotion regulation [27,28]. More specifically, studies have shown that emotion regulation acts as a mechanism that links self-compassion to either positive or negative mental health outcomes [27]. In the present study, it might be that lower levels of self-compassion negatively impacted men’s abilities to cope with stressful pandemic events through the effective use of emotion management strategies. This might help explain, at least in part, the association between low self-compassion and greater perceived stress. This finding also seems to be supported by other studies in which self-compassion was associated with better coping strategies and less stress [48]. The influences of sociodemographic and psychological factors (i.e., indicators of CMD) on the relationship between self-compassion and perceived stress will be further explored in the following sections.

Gender issues were also considered in the present study. For instance, non-cisgender men were more likely to be affected by mental health problems during the pandemic. It is known that men with non-heterosexual sexual orientations are more exposed to stressful events. They are significantly affected by “minority stress,” in which sexual identity becomes the target of stigma, discrimination, prejudice, and violence—i.e., LGBT phobias. In the context of the COVID-19 pandemic, the scientific literature has pointed out decreases in the psychosocial well-being of gay, bisexual, and transgender men [1,9]. The fact that they declare themselves to be non-cisgender and have low self-compassion and high perceived stress confirms the relevance of further discussions about homo-transphobic behaviors with an extensive trail of violence. This issue persists in society and exposes men who declare themselves non-heterosexual [49]. 

The association between low self-compassion and high perceived stress was statistically significant in the marital status category: without a partner. The experience of these crises sheds light on the relevance of a support network in minimizing the adverse effects that result from the uncertainties that these moments produce [50,51]. The presence of a partner for mutual support can represent a sense of relief and feelings that, in the event of more significant difficulties, someone can offer the strength and courage necessary to overcome challenges. On the other hand, experiencing crises in isolation increases feelings of insecurity, anxiety, and fear, reinforcing the situation’s perceived stress. Our data show that the relationship between low self-compassion and perceived stress was more significant among men without a partner, strengthening the hypothesis that isolation in crises amplifies the adverse effects on people’s emotions, making them more vulnerable.

In the final analysis model (multivariate model), only two factors were statistically associated with a combined situation of high perceived stress and low self-compassion: living with a friend and having CMD. The fact that men do not live with family members but with other colleagues generates a high burden of responsibilities and subsistence burdens that may be related to this result [28]. 

Furthermore, the results showed a robust association between CMD and a combined situation of high stress and low self-compassion (an association almost three times higher among those with a disorder). The two events that make up the outcome of interest (stress and self-compassion) refer to conditions/characteristics that the literature consistently associates with mental health. Self-compassion is a factor that can protect mental health [27,28] and stress; however, when constantly maintained at high levels, it poses a risk to mental functioning, producing suffering and illness [52,53]. Therefore, this result strengthens the hypothesis of the co-occurrence of unfavorable emotional situations and mental disorders. Based on this, it is possible to assume that actions that support self-compassion strategies and reduce stress levels will positively affect mental health, thereby reducing the occurrence of CMD.

On the other hand, measures to protect mental health may, in turn, also reduce high levels of stress and low self-compassion. Thus, coping strategies in any of these directions will amplify beneficial effects. They may act on the outcome of a specific situation as a primary objective. However, it still has the potential to reduce other adverse consequences as a secondary action—which is desirable. This aspect is of great interest for defining effective public health actions. Therefore, this result, particularly, has great utility and practical advantages for health policies.

Among the possible limitations of this research, the sample composition is an aspect that needs to be considered in analyzing our results. As we did not carry out sampling by probabilistic selection (it is a sample of volunteers—a convenience sample), the possibility that the studied group is not representative of the population of Brazilian men cannot be ruled out. In this sense, the sample’s particularities must be considered to avoid undue generalizations. It consists predominantly of men residing in urban areas with high schooling, access to the internet, and relatively good housing conditions. Therefore, the results obtained apply more directly and consistently to groups with these characteristics. Despite possible limitations in how the sample was selected, the sample we studied represents the characteristics of a portion of the Brazilian male population; thus, it offers relevant information that expresses the specific context experienced by this population group. 

More specifically, the results provide significant evidence about how Brazilian men perceived stress and practiced self-compassion during the major health crisis caused by the COVID-19 pandemic. Although there may be variations related to regional differences in Brazil, we believe that the results obtained in this study highlight a national trend and, therefore, reinforce the importance of considering individual and contextual factors in the promotion of men’s mental health in national public health care policies for this population.

## 5. Conclusions

We found that men with low levels of self-compassion reported higher levels of perceived stress; however, this association was moderated by intrapsychic and social aspects. For example, psychosocial variables, such as indicators of concomitant CMD, made this association stronger. In short, these results reveal that the relationship between self-compassion and perceived stress, in a pandemic context, needs to be understood from an intersectional or multidimensional perspective to capture its complexity adequately.

Our results can be useful in elaborating policies, programs, and strategies to strengthen and promote men’s mental health because they allow a better understanding of the relationship between individual and contextual characteristics. For example, the reviewed literature and the results of the present study highlight how self-compassion can inform mental health intervention programs and act as a protective factor. However, such interventions, to achieve greater levels of effectiveness, need to consider the degree of social support and the multiple indicators of men’s psychological distress.

## Figures and Tables

**Table 1 ijerph-19-08159-t001:** Distribution of men according to sociodemographic characteristics and mental health status in Brazilian men. Brazil, 2020.

Variables	*n*	%
**Sociodemographic Characteristics**		
Sexual identity (*n* = 945)		
Heterosexual	434	45.9
Non-heterosexual	511	54.1
Gender (*n* = 1006)		
Cisgender	938	93.2
Non-cisgender	68	6.8
Age group (*n* = 1006)		
18 to 39 years old	768	76.3
40 to 59 years old	213	21.2
60 years or older	25	2.5
Education (*n* = 1006)		
Elementary/high school	264	26.2
University education	742	73.8
Race/skin color (*n* = 1001)		
Not black	805	80.4
Black	196	19.6
Marital status (*n* = 1006)		
With partner	329	32.7
No partner	677	67.3
Monthly income (*n* = 1006)		
Up to US$ 859,56 *	646	64.2
US$1.074,45 ** or more	360	35.8
Occupation (*n* = 1006)		
Formal worker/stable income	536	53.3
Informal worker	254	25.2
Does not work	216	21.5
Housemate (*n* = 1005)		
Family	702	69.8
Friend(s)	68	6.8
Alone	235	23.4
Use of the health system (*n* = 1006)		
Exclusively the private system	261	25.9
Predominantly the SUS	745	74.1
**Self-compassion and perceived stress**		
No perceived stress or low self-compassion	196	19.6
Only high perceived stress	290	28.9
Only low self-compassion	27	2.7
Combined situation (high perceived stress and low self-compassion)	489	48.6
**Mental health status**		
CMD (*n* = 1006)		
Yes	460	45.7
No	546	54.3

SUS: Unified Health System. * Up to four Brazilian minimum wages. ** Five Brazilian minimum wages. NOTE: The total No. of men studied was 1006. The No. for each variable differed as a function of losses that ranged from 0.4% to 6.0% for sexual identity.

**Table 2 ijerph-19-08159-t002:** Estimates of the combined situation of low self-compassion and high perceived stress according to sociodemographic characteristics and mental health status Brazil, 2020. (*n* = 1006).

Variables	No	*p* (%)	*p*-Value *	PR Crude	95% CI
**High Perceived Stress and Low Self-Compassion**	489	48.6	-	-	-
**Sociodemographic Characteristics**					
Sexual identity (*n* = 945)					
Heterosexual	207	48.0	0.634	1.00	-
Non-heterosexual	237	46.5		0.97	0.85–1.11
Gender					
Cisgender	446	47.8	0.014	1.00	-
Non-cisgender	43	63.2		1.32	1.09–1.61
Age group					
18 to 39 years old	398	52.1	0.001	1.30	0.80–1.58
40 to 59 years old	81	38.0		0.95	0.57–1.58
60 years or older	10	40.0		1.00	-
Education					
Elementary/high school	118	44.9	0.137	0.89	0.77–1.04
University education	371	50.2		1.00	-
Race/skin color (*n* = 1001)					
Not black	386	48.2	0.334	1.00	-
Black	102	52.0		1.08	0.93–1.26
Marital status					
With partner	140	42.6	0.006	1.00	-
No partner	349	51.9		1.22	1.05–1.41
Monthly income					
Up to 4 minimum wages	299	46.6	0.059	1.13	1.00–1.29
5 minimum wages or more	190	52.8		1.00	-
Occupation					
Formal worker/stable income	271	50.7	0.239	1.00	-
informal worker	112	44.3		0.87	0.74–1.03
Does not work	106	49.5		0.98	0.83–1.15
Housemate (*n* = 1005)					
Family	324	46.2	0.037	1.00	-
Friend(s)	40	58.8		1.27	1.02–1.58
Alone	124	53.5		1.16	1.00–1.33
**Mental health status**					
CMD					
Yes	113	24.6	<0.001	1.00	-
No	376	69.4		2.82	2.38–3.35

* *p*-value obtained by Pearson’s Chi-Square test. *p*: Prevalence; PR: prevalence ratio; 95% CI: 95% Confidence Interval.

**Table 3 ijerph-19-08159-t003:** Multivariate model of factors associated with the interaction of low self-compassion and high perceived stress in Brazilian men. Brazil, 2020.

Variables	PR Ajustada	95% CI
Housemate		
Family	1.00	-
Friend(s)	1.26	1.03–1.53
Alone	1.11	0.97–1.26
CMD		
Yes	1.00	-
No	2.83	2.39–3.36

PR: prevalence ratio; 95%CI: Confidence Interval of 95%; AIC: Akaike Information Criterion; VIF: Variance Inflation Factor.

## Data Availability

The dataset generated during the current study are not publicly available but are available from the corresponding author (ARS) on reasonable request.

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
