# Peer review of "Influence of Sociodemographic and Emotional Factors on the Relationship between Self-Compassion and Perceived Stress among Men Residing in Brazil during the COVID-19 Pandemic"

_ijerph, 2022, doi:10.3390/ijerph19138159_

Round 1

Author Response

Reviewer:

Dear reviewer, thank you so much for the amazing suggestions.
Below are the answers to your questions:

1) Thank you so much for this great comment! Our study did well at the beginning of the pandemic. Little or nothing had been produced on the subject and there were no validated scales for Brazil. The authors wanted, in this way, to understand the phenomenon, to understand it. Hence the use of this method and not the use of scales.

2) Thank you so much for this great comment!
We make the gaps in the literature more evident by demonstrating that most studies have emphasized the relationships between self-compassion and psychological distress without considering the role of intermediate variables, such as sociodemographic and emotional variables. In this study, we demonstrate how these compromise variables this relationship, thus providing a more complex framework for understanding the effects of self-compassion on mental health.

2) We indicate the exact dates on which the research was carried out and the epidemiological situation in this period in Brazil. We discussed in the method how this affected the results;

3) It was a translation error, we fixed it.

4) The Kolmogorov-Smirnov test was used to assess the normality of the distribution of variables.

5) The authors detailed the cut-off points for the implemented scale that was used in the study.

6) The authors did not divide participants by contact with COVID-19 due to their occupation.

7) The phrase was deleted as it did not meet the new context.

8) Thank you very much for this comment. We add that: Considering that Brazil is a continental country with great cultural differences, it is impossible not to consider that the place of residence can influence the mechanisms available and used to cope with stress, as well as self-compassion.

Reviewer 2 Report

Thank you for the opportunity to review this interesting, timely, and important paper.

Overall, I think that the paper represents publishable research and analysis, but will need significant revisions to improve the paper before I recommend publication.

My major issues are the lack of a clearly defined gap in the literature, and major omissions of key concepts and ideas in the introduction, especially a number of critically important ideas and concepts related to the community being researched, the topic, and concepts that appear in the results and discussion. A more focused and logically organized introduction is necessary to present a coherent and compelling story. I recommend revision because I believe that the story is there to tell, but the paper is not currently successful in communicating that story to the reader.

The description of methods, especially with regards to the measures used, as well as other details, must be improved. The results as currently presented are not well-organized and presented, and there are some significant problems with tables. I recommend a complete revision of the results in order to focus on only those findings which are necessary for the current analysis. Some of the findings being reported do not seem to fit with the overall thesis or purpose for this analysis. Please review this section closely to determine which parts of the analysis the authors want to highlight. Sometimes this can mean not reporting an interesting finding, merely because it doesn't really fit with the core concept being presented. Better to lose that result than to muddy the waters.

The discussion also has major organization and clarity issues, as well as issues in the focus of the discussion. The approach I recommend is to present a clear summary of the entirety of the key findings in the first paragraph, and then move through each of the specific findings that are reported in the results in the format: Present a single key finding (We found that SES was correlated with Y). Discuss how this is related to the prior literature on SES and Y. (This is supported by the literature on SES, such as the recent paper Z by AAAAA et al.) And to especially highlight any of your findings that are novel. (Our findings on SES and Y are novel findings which do...)

I would also caution the authors about introducing new concepts in the discussion section which are not presented in the introduction. I recommend also either balancing the discussion of findings so that all findings are discussed in relatively the same number of sentences, or being more attentive to identifying the most important findings and then organizing this section around those findings. That will make it easier to devote more space to more important findings, but in a logical and clear fashion that is easier for readers to follow.

Finally, I recommend attention to how the authors discuss the ultimate purpose of this study related to interventions and/or policy. I recommend either removing this material or committing more fully to it.

I have attached a PDF draft of the article with line by line comments highlighted.

Thank you.

Author Response

Reviewer:
My main issues are the lack of a clearly defined gap in the literature, and major omissions of key concepts and ideas in the introduction, especially a number of critically important ideas and concepts related to the community research, topic, and concepts that appear in the results. and in the discussion. A more focused and logically organized introduction is needed to present a coherent and compelling story. I recommend proofreading because I believe the story is there to tell, but the newspaper is not successful in communicating that story to the reader.

Answer: We agree with the reviewer's excellent analysis and reorganized the manuscript in terms of presenting the most important variables through a reanalysis of the proposed theoretical-empirical model. Thus, the variables social support and intolerance to uncertainty were excluded, maintaining the main sociodemographic variables and indicators of common mental disorders. This reorganization of the model made the contributions of the study more evident, demonstrating the protective potential of self-compassion against the deleterious effects on men's mental health.
We also make the gaps in the literature more evident by demonstrating that most studies have emphasized the relationships between self-compassion and psychological distress without considering the role of intermediate variables, such as sociodemographic and emotional variables. . In this study, we demonstrate how these variables compromise this relationship, thus providing a more complex framework for understanding the effects of self-compassion on mental health.

Reviewer: The description of the methods, especially with regard to the measures used, as well as other details, should be improved. The currently presented results are not well organized and presented, and there are some significant problems with the tables. I recommend a thorough review of the results, in order to focus only on the findings necessary for the current analysis. Some of the reported findings do not seem to fit the general thesis or purpose of this analysis. Please review this section closely to determine which parts of the analysis the authors wish to highlight. Sometimes this can mean not reporting an interesting finding, simply because it doesn't really fit in with the core concept being presented. It is better to miss this result than to muddy the waters.

Answer: Method and results were carefully reviewed, highlighting the instruments used in data collection, the procedures, the characteristics of the participants. The results now make the relationships found more evident.

Reviewer: The discussion also has major issues of organization and clarity, as well as issues in the focus of the discussion. The approach I recommend is to present a clear summary of the entirety of the main findings from the first paragraph and then go through each of the specific findings that are reported in the results in the format: Present a single key finding (we found that SES was correlated with Y). Discuss how this relates to previous literature on SES and Y. (This is supported by literature on SES such as the recent article Z by AAAAA et al.) And to especially highlight any of your findings that are new. (Our findings about SES and Y are new discoveries that make...)

Answer: The discussion has been reorganized and begins with a brief summary of the main findings. Such findings are later interpreted in the light of the literature, highlighting the possible explanatory mechanisms of the relationships found.

Reviewer: I would also like to caution authors about introducing new concepts in the discussion section that are not presented in the introduction. I also recommend balancing the discussion of findings so that all findings are discussed in relatively the same number of sentences, or being more attentive to identifying the most important findings and then organizing this section around those findings. This will make it easier to dedicate more space to more important findings, but in a logical and clear way that is easier for readers to follow.

Answer: Only concepts and terms that are directly related to the discussion of the results found were kept. We believe that there was a significant improvement in the interpretation of the findings.

New analyzes were carried out (bivariate and multivariate), as well as a general review of the results, in order to focus only on the most relevant factors in determining self-compassion and perceived stress.

Round 2

Reviewer 2 Report

Thank you for the opportunity to review this revised manuscript. I found the authors revisions to be excellent improvements to the quality of the paper. Revisions made have also improved the value of the paper to the literature by clarifying and focusing the most important findings of the study presented. There are still some issues that I would like the authors to correct, but I applaud the authors for the incredible work that they have done in improving the draft. I did not find a response to reviewers document that outlines the revisions made in the manuscript, so my apologies if some of the below was mentioned with a rationale for retaining the original version.

Methods:

1. 2.2, the description of the sample provided here needs to clarification. The authors state that "the sample was estimated at 923 participants." Was this just for the purposes of determining the power, CI, etc.? Please include the final number of participants included in the analytical sample, and please describe the numbers of individuals who had to be excluded from the sample. Did the n=1,006 (pulled from the tables) represent every individual who provided consent? Were any responses deemed ineligible for the study (beyond the NB participants mentioned)? It is clear that not all of the n=1,006 completed the entire questionnaire, including race and sexual identity, were any potential participants in the study sample excluded for not completing the questionnaire? A flow-chart describing the total number of responses and including each of the excluded answers would be very helpful here. The authors provide an incomplete description in 2.5, this information could be included there, how many incomplete (and what criteria? how 'incomplete' before exclusion)? how many repeat/duplicates were excluded? were any excluded due to not meeting the age or geographical criteria described?

2. I would like more description on the measures used, what questions were used to collect the sociodemographics, As in my earlier review, I want the authors to include something about the "race/skin color" demographic variable. This is an appropriate variable for the social and cultural context of Brazil, but even a short description of the measures stating: "Brazilian conceptions of social identity based on racial characteristics, such as skin color, were used for sociodemographic measures." and then a sentence with a citation about the current common or primary race/skin color designations. I would also like the authors to describe how this was collected, i.e. were participants asked, "How would you describe your race?" or something else? Based on line 154, it seems that participants may have been provided a list, but more information would be helpful for me, as a reader, to gauge the quality of the race demographic, especially since this ends up not being presented as a key factor in the analysis. Since race/skin color was not shown as important, this needs to be described so that it is clear that the authors have not collapsed this category in order to shape the results in some way. A justification for this should be provided, was this category collapsed for a specific analytical reason? If there was a low n= for many categories, that is still important information about this sample (for example, if there were a very low number of people who identify as indigenous, then that may limit the generalizability of the study). I find the same criticism in the collapsing of all non-Heterosexual identifications, so some more description of what participants were asked, if multiple answers were allowed, was there an open-ended option or a none of the above? Were there low numbers of participants identifying as particular categories?

CRITICAL: I am now seeing a major issue with the term “transsexual” being included as a “sexual identity” when this term is highly problematic, and (as far as I am aware) “transsexual” is understood as being related to the gender of a person (i.e. someone could be a trans man, being assigned male at birth, but whose gender is man). This term is not used to identify someone’s sexual identity. I am (very vaguely) aware that there are gender and sexual orientations that are specific to Brazil which the authors may be referring to, if that is the case it is ESSENTIAL that the authors describe their measures and how these demographic variables were collected (i.e. using what question and format?), but also providing at least 1 sentence of context with at least one citation to direct those of us without specific expertise or experience in Brazil to more information that will improve our ability to relate these findings to non-Brazilian contexts.

3. The paragraph on lines 201-207 still needs clarification. The authors still need to describe more clearly the "social network" that was developed to "get closer to the participants." Please answer the following questions or provide the following information:

*The authors described this as "planned" but do not describe whether or not this social network was implemented, at what point in the research, for how long participants were involved (if this network was actually deployed), and other critical details, such as, how many participants in the study sample were involved (all? n=10? some other number?)

*What was the goal of this social network? There are several disparate goals mentioned, but it is unclear what the actual purpose this network served in the collection of data, the quality of the data collected, or in its analysis. Was this to build rapport? ("get closer to the participants") Was this part of the recruitment strategy? Was this network used for research dissemination, or intended for this goal? It is quite possible that all of these were intended, if so a more clear statement that researchers built a "social network" as part of the data collection that had X, Y, and Z goal, should be included.

*More description of the "social network" is required. Who built this, using what technology/software/platform? Was this created by one of the authors, another member of the study team? What expertise did they have in doing so? Was this "network" deployed on an existing social media architecture (such as a private or hidden group or similar)? 

*How was participant confidentiality protected from other participants and from possible outside intrusion into this "social network?" Were all participants involved in this "social network?" It is noted above that the questionnaire was built using Google Forms, but was this social network primarily used to distribute links and recruit participants? What do the authors mean by "personalized layout of the research" and what is a "digital animation" that was used to send the invitation? 

*Was data collected from interactions with participants on this social network? How did researchers on the team interact with participants on this network? What tools were put in place to protect participants from negative interactions or mental harm from their participation, was their moderation or other involvement to ensure that interactions using this network conformed to ethical obligations to protect study participants?

*Were participants who joined the social network provided informed consent about their participation? At what point were they provided this information and allowed to ask questions about their participation? How was consent recorded for participation in the social network?

Results:

4. I found some more inconsistencies in the graphs, although I find the current versions to be a significant improvement. There are still capitalization inconsistencies ("not black" should be "Not Black" & "non-cisgender" should be "Non-cisgender" & "informal worker" should be "Informal worker"). There are now inconsistencies in the use of the "," vs. the "." for decimals. Either would be acceptable for an international journal, but Table 2 now includes a line where the "," is used. 

5. In the tables the following categories are used in the "Occupation" demographic category includes "up to 4 minimum wages" and "5 minimum wages or more," but I am unclear as to what this means. Could the authors clarify whether this means 4 minimum wage positions and 5+ minimum wage positions held concurrently? If so, could the authors clarify what value the distinction between the 4 and 5 cutoff? From my perspective holding 2 or more minimum wage positions would qualify as being a person who is in extreme precarity, but that may not be the case in Brazil. This could be included in the expanded description of the sociodemographic measures above.

Results:

6. The authors need to include a paragraph presenting their sample and comparing it to the demographics of Brazilian men as a whole. I highly suggest this to be included as a new first paragraph of the results section. The authors do not note generalizability in their limitations, but it is standard to make some statement of how generalizable the sample is through this paragraph, which is then part of the limitations paragraph. I would have to guess that this sample is more educated, more “non-Black,” and perhaps more “Non-heterosexual,” than the general population of Brazilian men, but without that context it requires your reader to do the legwork to determine how they might use this study, and the confidence that they can use the author’s findings in future work.

Discussion:

7. Some of the new material includes poorly formatted or highly garbled citations in the new version. This may be the case elsewhere, and I recommend checking/correcting all in-line parenthetical citations. For example, please refer to line 296-297: “For instance, non-cisgender men were more likely et al., 2021). 2019). to be affected by…,” line 324, and line 330: “…the protection of mental health (In-wood & Ferrari, 2018). ; Neff & Germer, 2017) and…”

8. Finally, I believe that the authors need to include a discussion of the generalizability of their findings based on how representative this sample is as a whole, if the demographics of this sample diverge from Brazilian men as a whole, it is critical that the authors provide the reason that their findings are still important even if limited in this specific way. The authors indicate that their sample is geographically biased, but they do not report geographical location, please include this along with the other demographic measures, especially as the exclusion of this data weakens the utility and value of the study. If, for example, more urban participants are included, the study is more useful when reporting that than when that data is purposefully not reported. When we lack that data it makes the rest of the dataset more suspect, and thus damages even the value of what is reported because we can’t even see how biased the participant distribution is, etc.

Thank you once again for the chance to review. I found this version to be a significant improvement, and, overall, a joy to read.

Author Response

Revisões IJERPH – Minor

Dear editor,

We are very honored and satisfied with the investment and work dedicated to our manuscript. The reviewers' assessments were thorough, detailed and enriched. We consider that our manuscript has been strengthened and increased scientific rigor and methodological quality.

We inform you that we accept the considerations in their entirety. What was not incorporated into the text is justified in this opinion.

Reviewers' considerations:

Methods:

  1. the sample description provided here needs clarification. The authors state that "the sample was estimated at 923 participants". Was this just to determine wattage, CI, etc.? Please include the final number of participants included in the analytical sample and describe the number of individuals who had to be excluded from the sample. Did n=1,006 (extracted from the tables) represent all individuals who provided consent? Were any responses considered ineligible for the study (other than the RN participants mentioned)? Of course, not all of the n=1,006 completed the entire questionnaire, including race and sexual identity, were any potential participants in the study sample excluded for not completing the questionnaire? A flowchart describing the total number of responses and including each of the excluded responses would be very helpful here. The authors provide an incomplete description in 2.5, could this information be included there, how many incomplete (and what criteria? how 'incomplete' before exclusion)? how many repeats/duplicates were excluded? were any excluded for not meeting the age or geographic criteria described?

Authors' response:

Detailed information about the sample, losses, and defined criteria are highlighted in red in the text. Thus, the descriptions were more detailed in the text.

Reviewers' considerations:

  1. I would like more description about the measures used, which questions were used to collect the sociodemographic data. As with my previous review, I want the authors to include something about the "race/skin color" demographic variable. This is an appropriate variable for Brazil's social and cultural context, but even a brief description of the measures states: "Brazilian conceptions of social identity based on racial characteristics, such as skin color, were used for sociodemographic measures". and then a sentence with a quote about current common or primary race/skin color designations. I would also like the authors to describe how this was collected, ie, participants asked, "How would you describe your race?" or something else? Based on row 154, it appears that the participants may have been given a list, but more information would be helpful for me as a reader to gauge the quality of the race demographics, especially as this ends up not being presented as a key factor in the analysis. As race/skin color did not prove to be important, this needs to be described so that it is clear that the authors did not collapse this category to shape the results in any way. A justification for this must be provided, was this category collected for a specific analytical reason? If there was a low n= for many categories, it is still important information about this sample (eg, if there were a very low number of people who identify as indigenous, this may limit the generalizability of the study). I find the same criticism in the breakdown of all non-heterosexual identifications, so a little more description than participants were asked, if multiple responses were allowed, was there an option open or none of the above? Was there a low number of participants who identified themselves as specific categories?

CRITIC: I am now seeing a major problem with the term “transsexual” being included as “sexual identity” when that term is highly problematic and (as far as I know) “transsexual” is understood to relate to a person’s gender (i.e., someone can be a trans man, designated male at birth, but whose gender is male). This term is not used to identify someone's sexual identity. I am (very vaguely) aware that there are Brazil-specific sexual and gender orientations that the authors may be referring to, if any, it is ESSENTIAL that the authors describe their measurements and how these demographic variables were collected (i.e., using which question and format?), but also providing at least 1 sentence of context with at least one citation to direct those of us with no specific knowledge or experience in Brazil to more information that will improve our ability to relate these findings to non-Brazilian contexts.

Authors' response:

The explanatory detail on the race/color variable was described in the text, emphasizing the specific characteristics of Brazil and how the variable was collected and considered in our study. The added data were supported by citations from previously published studies. In addition, we brought information about the inclusion of men with a self-reported indigenous race/color, considering the limitations in reaching this population. The changes are detailed in the red text.

We clarify how the category of sexual orientation and gender identity was measured. We apologize for the mistake in the use of the term sexual identity and the use of the transsexual concept. This was a translation error. We detailed how we collected the data and how it was accommodated in our study.

Reviewers' considerations:

  1. The paragraph of lines 201-207 still needs clarification. The authors still need to more clearly describe the “social network” that was developed to “get closer to the participants”. Please answer the following questions or provide the following information:

*The authors described this as "planned" but did not describe whether or not this social network was implemented, at what point in the research, how long participants were involved (if this network was actually implemented), and other critical details such as how, how many participants in the study sample were involved (all? n=10? any other number?)

- Answer: The social network was used as a reliable, public and unique place where the link could be accessible to all participants; and could also be used by participants to send to other colleagues, friends and partners. This page was created by marketing, design, and communication professionals and is used for recruitment, approaching research participants, disseminating research results, and posting guidelines for health promotion.

*What was the purpose of this social network? There are several disparate objectives mentioned, but it is not clear what the real purpose of this network is in data collection, in the quality of the data collected or in its analysis. Was this to build relationship? ("Get closer to the participants") Was this part of the recruiting strategy? Was this network used to disseminate research or intended for this purpose? It is quite possible that this was all intended, if so, a clearer claim that the researchers built a "social network" as part of data collection that aimed at X, Y and Z should be included.

- Answer: - Thank you very much for this comment. We seek to make this information clearer, as follows: “This page was created by marketing, design, and communication professionals and is used for recruitment, approaching research participants, disseminating research results, and posting guidelines for health promotion.  All contacts with participants were made individually to ensure participants' privacy and confidentiality of the data collected, in compliance with the general law of data protection and the resolution on ethics in research in a virtual environment”.

*More description of "social network" is required. Who built this, using which technology/software/platform? Was this created by one of the authors, another member of the study team? What experience did they have to do this? Was this "network" deployed on an existing social media architecture (like a private or hidden group or similar)?

- Answer: - It was in fact an already known and published social network, instagram. Creation details have been added.

*How was the confidentiality of participants protected from other participants and from possible external intrusion into this "social network"? Were all participants involved in this "social network"? It is noted above that the questionnaire was built using Google Forms, but was this social network used primarily to distribute links and recruit participants? What do the authors understand by "custom survey layout" and what is a "digital animation" that was used to send the invitation?

- Answer: - The reviewer confused some terms in this case. The social network page was used only as a place to access the link to the form that was available on google forms. Also, as it is a page, some useful information for the health of the participants (such as information on mental health services) was available.

*Was the data collected from interactions with participants in this social network? How did the team's researchers interact with the participants of this network? What tools were put in place to protect participants from negative interactions or mental harm from their participation, moderation, or other involvement to ensure that interactions using this network complied with ethical obligations to protect study participants? *Did participants who joined the social network provide informed consent about their participation? At what point did they receive this information and were allowed to ask questions about their participation? How was consent to participate in the social network registered?

- Answer: -There were no interactions in the social network between the researchers and the participants, nor was information collected. It was not necessary to protect the participants' information as the social network is public.

This information is available here: “In order to get closer to the participants, a planned data collection strategy was developed, namely: the use of a social network linked to the research group that conducted the study, through a page on Instagram named @cuidadoasaudehomens, elaboration of a personalized layout of the research, creation of digital animation to send the invitation and interaction with participants in the virtual environment, holding meetings, conversation circles, and presentation of the research in virtual spaces of male socialization, as a way of propagating the research in different areas and capturing other groups of men residing in the country. This page was created by marketing, design, and communication professionals and is used for recruitment, approaching research participants, disseminating research results, and posting guidelines for health promotion.

All contacts with participants were made individually to ensure participants' privacy and confidentiality of the data collected, in compliance with the general law of data protection and the resolution on ethics in research in a virtual environment.

The research used a self-administered, semi-structured electronic form composed of structured questions related to sociodemographic, occupational, and health characteristics and aimed at experiencing the pandemic. Open questions were also inserted; however, they were not incorporated in this study. The Google Forms® platform was used, which is free to use, as it is a widely known platform in Brazil, easy to access and use, using resources to ensure data security and protection, such as encryption.”

Reviewers' considerations:

Results:

  1. I found some more inconsistencies in the graphics, although I think the current versions are a significant improvement. There are still capitalization inconsistencies ("non-black" must be "non-black" and "non-cisgender" must be "non-cisgender" and "informal worker" must be "informal worker"). There are now inconsistencies in the use of "," vs. O "." to decimals. Either would be acceptable for an international journal, but Table 2 now includes a line where "," is used.

Authors' response:

Percentage data has been corrected.

Reviewers' considerations:

  1. In the tables, the following categories are used in the "Occupation" demographic category includes "up to 4 minimum wages" and "5 minimum wages or more", but I am not clear what this means. Could the authors clarify whether this means 4 minimum-wage positions and 5 or more minimum-wage positions held concurrently? If so, could the authors clarify the value of the distinction between cut-off points 4 and 5? From my point of view, holding 2 or more minimum wage positions would qualify as a person in extreme precariousness, but this may not be the case in Brazil. This could be included in the expanded description of sociodemographic measures above.

- Answer: Thank you so much for this important and relevant comment! In this case, we transform minimum wages into the amounts received in dollars. Thus, readers can assess GDP or Purchasing Power Comparative - PPC and make relevant analyses.

The contributions qualified our manuscript and were essential to repair the imperfections, making the replication capacity more adequate.

Results:

  1. Authors should include a paragraph presenting their sample and comparing it to the demographics of Brazilian men as a whole. I suggest that this be included as a new first paragraph in the results section. Authors do not note generalizability in their limitations, but it is standard to make some statement of how generalizable the sample is through this paragraph, which is then part of the limitations paragraph. I would have to guess that this sample is more educated, more “non-black” and perhaps more “non-heterosexual” than the general population of Brazilian men, but without that context, it requires your reader to do the legwork to determine how they can use this study and confidence that they can use the author's findings in future work.

Authors' response:

The suggested paragraph has been added and is highlighted in the text.

Reviewers' considerations:

Discussion:

  1. Some of the new material includes poorly formatted or highly distorted citations in the new version. This may be the case elsewhere, and I recommend checking/correcting all citations in parentheses inline. For example, see line 296-297: “For example, non-cisgender men were more likely et al., 2021). 2019). be affected by…”, line 324, and line 330: “…the protection of mental health (In-wood & Ferrari, 2018). ; Neff & Germer, 2017) and…”

Authors' response:

Adjustments to the references were made and a discussion topic was added in relation to the socioeconomic variable.

Reviewers' considerations:

  1. Finally, I believe the authors need to include a discussion of generalizability of their findings based on the representativeness of this sample as a whole, if the demographics of this sample differ from Brazilian men as a whole, it is critical that the authors provide the reason that his findings are still important, even if limited in this specific way. The authors indicate that their sample is geographically biased, but they do not report geographic location, please include this along with the other demographic measures, especially as excluding this data weakens the usefulness and value of the study. If, for example, more urban participants are included, the study will be more helpful in reporting this than when these data are purposely not reported. When we don't have this data, the rest of the dataset becomes more suspect and therefore even harms the value of what is reported, because we can't even see how biased the distribution of participants is, etc.

Thanks again for the opportunity to rate. I found this version a significant improvement and overall a joy to read.

Authors' response:

Once again we thank you for your valuable contributions.

As requested, we revised the text and included paragraphs that express the generalization of the findings found, presenting the strengths and care that must be taken when consuming the evidence presented, considering the specificities of the representation of our sample, in comparison with the rest of the Brazilian male population. In addition, we include the reasons for our findings and their potential for contribution and reach.

We believe that the evaluation process has enhanced and strengthened the potential for reading and citing the manuscript, if accepted for publication. We reinforce that studies involving the male population in the pandemic are scarce in the world, especially in Latin America, which reinforces the need for their publication, considering the potential reach of the journal.

Agradecem,

Os autorres

Brazil, 2022.